# Approximation Bounds for Hierarchical Clustering: Average Linkage, Bisecting K-means, and Local Search

**Benjamin Moseley**[*]
Carnegie Mellon University
Pittsburgh, PA 15213, USA
moseleyb@andrew.cmu.edu

**Joshua R. Wang**[†]
Department of Computer Science Stanford University
353 Serra Mall, Stanford, CA 94305, USA
joshua.wang@cs.stanford.edu

## Abstract

Hierarchical clustering is a data analysis method that has been used for decades. Despite its widespread use, the method has an underdeveloped analytical foundation. Having a well understood foundation would both support the currently used methods and help guide future improvements. The goal of this paper is to give an analytic framework to better understand observations seen in practice. This paper considers the dual of a problem framework for hierarchical clustering introduced by Dasgupta [Das16]. The main result is that one of the most popular algorithms used in practice, average linkage agglomerative clustering, has a small constant approximation ratio for this objective. Furthermore, this paper establishes that using bisecting $k$-means divisive clustering has a very poor lower bound on its approximation ratio for the same objective. However, we show that there are divisive algorithms that perform well with respect to this objective by giving two constant approximation algorithms. This paper is some of the first work to establish guarantees on widely used hierarchical algorithms for a natural objective function. This objective and analysis give insight into what these popular algorithms are optimizing and when they will perform well.

## 1 Introduction

Hierarchical clustering is a widely used method to analyze data. See [MC12, KBXS12, HG05] for an overview and pointers to relevant work. In a typical hierarchical clustering problem, one is given a set of $n$ data points and a notion of similarity between the points. The output is a hierarchy of clusters of the input. Specifically, a dendrogram (tree) is constructed where the leaves correspond to the $n$ input data points and the root corresponds to a cluster containing all data points. Each internal node of the tree corresponds to a cluster of the data points in its subtree. The clusters (internal nodes) become more refined as the nodes are lower in the tree. The goal is to construct the tree so that the clusters deeper in the tree contain points that are relatively more similar.

There are many reasons for the popularity of hierarchical clustering, including that the number of clusters is not predetermined and that the clusters produced induce taxonomies that give meaningful ways to interpret data.

Methods used to perform hierarchical clustering are divided into two classes: agglomerative and divisive. **Agglomerative** algorithms are a bottom-up approach and are more commonly used than

---

[*]Benjamin Moseley was supported in part by a Google Research Award, a Yahoo Research Award and NSF Grants CCF-1617724, CCF-1733873 and CCF-1725661. This work was partially done while the author was working at Washington University in St. Louis.

[†]Joshua R. Wang was supported in part by NSF Grant CCF-1524062.

divisive approaches [HTF09]. In an agglomerative method, each of the $n$ input data points starts as a cluster. Then iteratively, pairs of similar clusters are merged according to some appropriate metric of similarity. Perhaps the most popular metric to define similarity is **average linkage** where the similarity between two clusters is defined as the average similarity between all pairs of data points in the two clusters. In average linkage agglomerative clustering the two clusters with the highest average similarity are merged at each step. Other metrics are also popular. Related examples include: **single linkage**, where the similarity between two clusters is the maximum similarity between any two single data points in each cluster, and **complete linkage**, where the distance is the minimum similarity between any two single data points in each cluster.

**Divisive** algorithms are a top-down approach where initially all data points belong to a single cluster. Splits are recursively performed, dividing a cluster into two clusters that will be further divided. The process continues until each cluster consists of a single data point. In each step of the algorithm, the data points are partitioned such that points in each cluster are more similar than points across clusters. There are several approaches to perform divisive clustering. One example is bisecting $k$-means where $k$-means is used at each step with $k = 2$. For details on bisecting $k$-means, see [Jai10].

**Motivation:** Hierarchical clustering has been used and studied for decades. There has been some work on theoretically quantifying the quality of the solutions produced by algorithms, such as [ABBL12, AB16, ZB09, BA08, Das16]. Much of this work focuses on deriving the structure of solutions created by algorithms or analytically describing desirable properties of a clustering algorithm. Though the area has been well-studied, there is no widely accepted formal problem framework. Hierarchical clustering describes a class of algorithmic methods rather than a problem with an objective function. Studying a formal objective for the problem could lead to the ability to objectively compare different methods; there is a desire for the community to investigate potential objectives. This would further support the use of current methods and guide the development of improvements.

This paper is concerned with investigating objectives for hierarchical clustering. The main goal and result of this paper is giving a natural objective that results in a theoretical guarantee for the most commonly used hierarchical clustering algorithm, average linkage agglomerative clustering. This guarantee gives support for why the algorithm is popular in practice and the objective gives insight into what the algorithm optimizes. This paper also proves a bad lower bound on bisecting $k$-means with respect to the same natural objective. This objective can therefore be used as a litmus test for the applicability of particular algorithms. This paper further gives top-down approaches that do have strong theoretical guarantees for the objective.

**Problem Formulation:** Towards this paper's goal, first a formal problem framework for hierarchical clustering needs to be established. Recently, Dasgupta [Das16] introduced a new problem framework for hierarchical clustering. This work justified their objective by establishing that for several sample problem instances, the resulting solution corresponds to what one might expect out of a desirable solution. This work has spurred considerable interest and there have been several follow up papers [CC17, Das16, RP16].

In the problem introduced by Dasgupta [Das16] there is a set of $n$ data points as input and for two points $i$ and $j$ there is a weight $w_{i,j}$ denoting their similarity. The higher the weight, the larger the similarity. This is represented as a weighted complete graph $G$. In the problem the output is a (full) *binary* tree where the leaves of the tree correspond to the input data points. For each pair of points $i$ and $j$, let $T[i \vee j]$ denote the subtree rooted at $i$ and $j$'s least common ancestor. Let $\texttt{leaves}(T[i \vee j])$ denote the set of leaves in the tree $T[i \vee j]$. The goal is to construct the tree such that the cost $\text{cost}_G(T) := \sum_{i,j \in [n]} w_{ij} |\texttt{leaves}(T[i \vee j])|$ is minimized. Intuitively, this objective enforces that more similar points $i$ and $j$ should have a lower common ancestor in the tree because the weight $w_{i,j}$ is large and having a smaller least common ancestor ensures that $|\texttt{leaves}(T[i \vee j])|$ is smaller. In particular, more similar points should be separated at lower levels of the hierarchical clustering.

For this objective, several approximation algorithms have been given [CC17, Das16, RP16]. It is known that there is a divisive clustering algorithm with an approximation ratio of $O(\sqrt{\log n})$ [CC17]. In particular, the algorithm gives a $O(\alpha_n)$-approximation where $\alpha_n$ is the approximation ratio of the sparsest cut subroutine [CC17]. Furthermore, assuming the Small-Set Expansion Hypothesis, every algorithm is a $\omega(1)$-approximation [CC17]. The current best known bound on $\alpha_n$ is $O(\sqrt{\log n})$ [ARV09]. Unfortunately, this conclusion misses one of our key goals in trying to establish an

objective function. While the algorithms and analysis are ingenious, none of the algorithms with theoretical guarantees are from the class of algorithms used in practice. Due to the complexity of the proposed algorithms, it will also be difficult to put them into practice.

Hence the question still looms: are there strong theoretical guarantees for practical algorithms? Is the objective from [Das16] the ideal objective for our goals? Is there a natural objective that admits solutions that are provably close to optimal?

**Results:** In this paper, we consider an objective function motivated by the objective introduced by Dasgupta in [Das16]. For a given tree $T$ let $|\texttt{non-leaves}(T[i \vee j])|$ be the total number of leaves that *are not* in the subtree rooted at the least common ancestor of $i$ and $j$. The objective in [Das16] focuses on constructing a binary tree $T$ to minimize the cost $\text{cost}_G(T) := \sum_{i,j \in [n]} w_{ij}|\texttt{leaves}(T[i \vee j])|$. This paper considers the *dual* problem where $T$ is constructed to maximize the revenue $\text{rev}_G(T) := \sum_{i,j \in [n]} w_{ij}|\texttt{non-leaves}(T[i \vee j])| = (n \sum_{i,j \in [n]} w_{i,j}) - \text{cost}_G(T)$. It is important to observe that the optimal clustering *is the same for both objectives*. Due to this, all the examples given in [Das16] motivating their objective by showing desirable structural properties of the optimal solution also apply to the objective considered in this paper. Our objective can be interpreted similarly to that in [Das16]. In particular, similar points $i$ and $j$ should be located lower in the tree as to maximize $|\texttt{non-leaves}(T[i \vee j])|$, the points that get separated at high levels of the hierarchical clustering.

This paper gives a thorough investigation of this new problem framework by analyzing several algorithms for the objective. The main result is establishing that average linkage clustering is a $\frac{1}{3}$-approximation. This result gives theoretical justification for the use of average linkage clustering and, additionally, this shows that the objective considered is tractable since it admits $\Omega(1)$-approximations. This suggests that the objective captures a component of what average linkage is optimizing for.

This paper then seeks to understand what other algorithms are good for this objective. In particular, is there a divisive algorithm with strong theoretical guarantees? What can be said about practical divisive algorithms? We establish that bisecting $k$-means is no better than a $O(\frac{1}{\sqrt{n}})$ approximation. This establishes that this method is *very* poor for the objective considered. This suggests that bisecting $k$-means is optimizing for something different than what average linkage optimizes for.

Given this negative result, we question whether there are divisive algorithms that optimize for our objective. We answer this question affirmatively by giving a local search strategy that obtains a $\frac{1}{3}$-approximation as well as showing that randomly partitioning is a tight $\frac{1}{3}$-approximation. The randomized algorithm can be found in the supplementary material.

**Other Related Work:** Very recently a contemporaneous paper [CKMM17] done independently has been published on ArXiv. This paper considers another class of objectives motivated by the work of [Das16]. For their objective, they also derive positive results for average linkage clustering.

## 2 Preliminaries

In this section, we give preliminaries including a formal definition of the problem considered and basic building blocks for later algorithm analysis.

In the **Revenue Hierarchical Clustering Problem** there are $n$ input data points given as a set $V$. There is a weight $w_{i,j} \geq 0$ between each pair of points $i$ and $j$ denoting their similarity, represented as a complete graph $G$. The output of the problem is a rooted tree $T$ where the leaves correspond to the data points and the internal nodes of the tree correspond to clusters of the points in the subtree. We will use the indices $1, 2, \ldots n$ to denote the leaves of the tree. For two leaves $i$ and $j$, let $T[i \vee j]$ denote the subtree rooted at the least common ancestor of $i$ and $j$ and let the set $\texttt{non-leaves}(T[i \vee j])$ denote the number of leaves in $T$ that are not in $T[i \vee j]$. The objective is to construct $T$ to maximize the revenue $\text{rev}_G(T) = \sum_{i \in [n]} \sum_{j \neq i \in [n]} w_{i,j}|\texttt{non-leaves}(T[i \vee j])|$.

We make no assumptions on the structure of the optimal tree $T$; however, one optimal tree is a binary tree, so we may restrict the solution to binary trees without loss of generality. To see this, let $\texttt{leaves}(T[i \vee j])$ be the set of leaves in $T[i \vee j]$ and $\text{cost}_G(T) := \sum_{i,j} w_{ij}|\texttt{leaves}(T[i \vee j])|$. The objective considered in [Das16] focuses on minimizing $\text{cost}_G(T)$. We note than $\text{cost}_G(T) + \text{rev}_G(T) = n \sum_{i,j} w_{i,j}$, so the optimal solution to minimizing $\text{cost}_G(T)$ is the same as the optimal

solution to maximizing $\text{rev}_G(T)$. In [Das16] it was shown that the optimal solution for any input is a binary tree.

As mentioned, there are two common types of algorithms for hierarchical clustering: agglomerative (bottom-up) algorithms and divisive (top-down) algorithms. In an agglomerative algorithm, each vertex $v \in V$ begins in separate cluster, and each iteration of the algorithm chooses two clusters to merge into one. In a divisive algorithm, all vertices $v \in V$ begin in a single cluster, and each iteration of the algorithm selects a cluster with more than one vertex and partitions it into two small clusters.

In this section, we present some basic techniques which we later use to analyze the effect each iteration has on the revenue. It will be convenient for us to think of the weight function as taking in two vertices instead of an edge, i.e. $w : V \times V \to \mathbb{R}^{\geq 0}$. This is without loss of generality, because we can always set the weight of any nonedge to zero (e.g. $w_{vv} = 0 \quad \forall v \in V$).

To bound the performance of an algorithm it suffices to bound $\text{rev}_G(T)$ and $\text{cost}_G(T)$ since $\text{rev}_G(T) + \text{cost}_G(T) = n\sum_{i,j} w_{i,j}$. Further, let $T^*$ denote the optimal hierarchical clustering. Then its revenue is at most $\text{rev}_G(T^*) \leq (n-2)\sum_{ij} w_{ij}$. This is because any edge $ij$ can have at most $(n-2)$ non-leaves for its subtree $T[i \vee j]$; $i$ and $j$ are always leaves.

## 2.1 Analyzing Agglomerative Algorithms

In this section, we discuss a method for bounding the performance of an agglomerative algorithm. When an agglomerative algorithm merges two clusters $A, B$, this determines the least common ancestor for any pair of nodes $i, j$ where $i \in A$ and $j \in B$. Given this, we define the revenue gain *due to* merging $A$ and $B$ as, $\text{merge-rev}_G(A, B) := (n - |A| - |B|)\sum_{a \in A, b \in B} w_{ab}$.

Notice that the final revenue $\text{rev}_G(T)$ is exactly the sum over iterations of the revenue gains, since each edge is counted exactly once: when its endpoints are merged into a single cluster. Hence, $\text{rev}_G(T) = \sum_{\text{merges } A, B} \text{merge-rev}_G(A, B)$.

We next define the *cost* of merging $A$ and $B$ as the following. This is the potential revenue lost by merging $A$ and $B$; revenue that can no longer be gained after $A$ and $B$ are merged, but was initially possible. Define, $\text{merge-cost}_G(A, B) := |B|\sum_{a \in A, c \in [n] \backslash (A \cup B)} w_{ac} + |A|\sum_{b \in B, c \in [n] \backslash (A \cup B)} w_{bc}$.

The total cost of the tree $T$, $\text{cost}_G(T)$, is exactly the sum over iterations of the cost increases, plus an additional $2\sum_{ij} w_{ij}$ term that accounts for each edge being counted towards its own endpoints. We can see why this is true if we consider a pair of vertices $i, j \in [n]$ in the final hierarchical clustering $T$. If at some point a cluster containing $i$ is merged with a third cluster before it gets merged with the cluster containing $j$, then the number of leaves in $T[i \vee j]$ goes up by the size of the third cluster. This is exactly the quantity captured by our cost increase definition. Aggregated over all pairs $i, j$ this is the following, $\text{cost}_G(T) = \sum_{i,j \in [n]} w_{ij} |\texttt{leaves}(T[i \vee j])| = 2\sum_{i,j \in [n]} w_{ij} + \sum_{\text{merges } A, B} \text{merge-cost}_G(A, B)$.

## 2.2 Analyzing Divisive Algorithms

Similar reasoning can be used for divisive algorithms. The following are revenue gain and cost increase definitions for when a divisive algorithm partitions a cluster into two clusters $A, B$. Define, $\text{split-rev}_G(A, B) := |B|\sum_{a,a' \in A} w_{aa'} + |A|\sum_{b,b' \in B} w_{bb'}$ and $\text{split-cost}_G(A, B) := (|A| + |B|)\sum_{a \in A, b \in B} w_{ab}$.

Consider the revenue gain. For $a, a' \in A$ we are now guaranteed that when the nodes in $B$ are split from $A$ then every node in $B$ will not be a leaf in $T[a \vee a']$ (and a symmetric term for when they are both in $B$). On the cost side, the term counts the cost of any pairs $a \in A$ and $b \in B$ that are now separated since we now know their subtree $T[i \vee j]$ has exactly the nodes in $A \cup B$ as leaves.

# 3 A Theoretical Guarantee for Average Linkage Agglomerative Clustering

In this section, we present the main result, a theoretical guarantee on average linkage clustering. We additionally give a bad example lower bounding the best performance of the algorithm. See [MC12] for details and background on this widely used algorithm. The formal definition of the algorithm

is given in the following pseudocode. The main idea is that initially all $n$ input points are in their own cluster and the algorithm recursively merges clusters until there is one cluster. In each step, the algorithm mergers the clusters $A$ and $B$ such that the pair maximizes the average distances of points between the two clusters, $\frac{1}{|A||B|}\sum_{a\in A,b\in B} w_{ab}$.

---

**Data:** Vertices $V$, weights $w : E \to \mathbb{R}^{\geq 0}$
Initialize clusters $\mathcal{C} \leftarrow \cup_{v\in V}\{v\}$;
**while** $|\mathcal{C}| \geq 2$ **do**
    Choose $A, B \in \mathcal{C}$ to maximize $\bar{w}(A,B) := \frac{1}{|A||B|}\sum_{a\in A,b\in B} w_{ab}$;
    Set $\mathcal{C} \leftarrow \mathcal{C} \cup \{A \cup B\} \setminus \{A,B\}$;
**end**

**Algorithm 1:** Average Linkage

---

The following theorem establishes that this algorithm is only a small constant factor away from optimal.

**Theorem 3.1.** *Consider a graph $G = (V,E)$ with nonnegative edge weights $w : E \to \mathbb{R}^{\geq 0}$. Let the hierarchical clustering $T^*$ be a optimal solution maximizing of $rev_G(\cdot)$ and let $T$ be the hierarchical clustering returned by Algorithm 1. Then, $rev_G(T) \geq \frac{1}{3}rev_G(T^*)$.*

*Proof.* Consider an iteration of Algorithm 1. Let the current clusters be in the set $\mathcal{C}$, and the algorithm chooses to merge clusters $A$ and $B$ from $\mathcal{C}$. When doing so, the algorithm attains a revenue gain of the following. Let $\bar{w}(A,B) = \frac{1}{|A||B|}\sum_{a\in A,b\in B} w_{ab}$ be the average weight of an edge between points in $A$ and $B$.

$$\text{merge-rev}_G(A,B) = (n - |A| - |B|)\sum_{a\in A,b\in B} w_{ab} = \sum_{C\in\mathcal{C}\setminus\{A,B\}}|C|\sum_{a\in A,b\in B} w_{ab}$$

$$= \sum_{C\in\mathcal{C}\setminus\{A,B\}}|C||A||B|\bar{w}(A,B)$$

while at the same time incurring a cost increase of:

$$\text{merge-cost}_G(A,B) = |B|\sum_{a\in A,c\in[n]\setminus(A\cup B)} w_{ac} + |A|\sum_{b\in B,c\in[n]\setminus(A\cup B)} w_{bc}$$

$$= |B|\sum_{C\in\mathcal{C}\setminus\{A,B\}}\sum_{a\in A,c\in C} w_{ac} + |A|\sum_{C\in\mathcal{C}\setminus\{A,B\}}\sum_{b\in B,c\in C} w_{bc}$$

$$= \sum_{C\in\mathcal{C}\setminus\{A,B\}}|B||A||C|\bar{w}(A,C) + \sum_{C\in\mathcal{C}\setminus\{A,B\}}|A||B||C|\bar{w}(B,C)$$

$$\leq \sum_{C\in\mathcal{C}\setminus\{A,B\}}|B||A||C|\bar{w}(A,B) + \sum_{C\in\mathcal{C}\setminus\{A,B\}}|A||B||C|\bar{w}(A,B)$$

$$= 2 \cdot \text{merge-rev}_G(A,B)$$

Intuitively, every time this algorithm loses two units of potential it cements the gain of one unit of potential, which is why it is a $\frac{1}{3}$-approximation. Formally:

$$\text{cost}_G(T) = 2\sum_{i,j} w_{ij} + \sum_{\text{merges } A, B}\text{merge-cost}_G(A,B) \leq 2\sum_{i,j} w_{ij} + 2\cdot\sum_{\text{merges } A, B}\text{merge-rev}_G(A,B)$$

$$\leq 2\sum_{i,j} w_{ij} + 2\cdot\text{rev}_G(T)$$

Now the revenue can be bounded as follows.

$$\text{rev}_G(T) \geq n\sum_{ij} w_{ij} - \text{cost}_G(T) \geq n\sum_{ij} w_{ij} - 2\sum_{i,j} w_{ij} - 2\cdot\text{rev}_G(T)$$

$$\text{rev}_G(T) \geq \frac{n-2}{3}\sum_{ij} w_{ij} \geq \frac{1}{3}\text{rev}_G(T^*)$$

where the last step follows from the fact that it is impossible to have more than $n - 2$ non-leaves. $\quad\square$

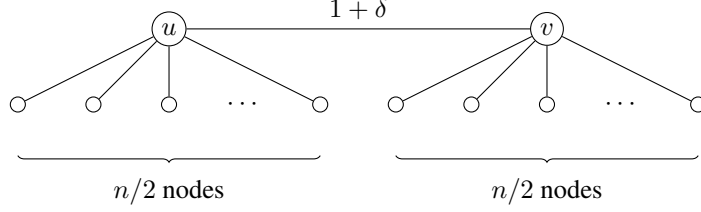

$1 + \delta$

$n/2$ nodes          $n/2$ nodes

Figure 1: Hard graph for Average Linkage ($k = 2$ case).

In the following, we establish that the algorithm is at best a $1/2$ approximation. The proof can be found in Section 1 of the supplementary material.

**Lemma 3.2.** *Let $\epsilon > 0$ be any fixed constant. There exists a graph $G = (V, E)$ with nonnegative edge weights $w : E \rightarrow \mathbb{R}^{\geq 0}$, such that if the hierarchical clustering $T^*$ is an optimal solution of $rev_G(\cdot)$ and $T$ is the hierarchical clustering returned by Average Linkage, $rev_G(T) \leq \left( \frac{1}{2} + \epsilon \right) rev_G(T^*)$.*

## 4   A Lower Bound on Bisecting $k$-means

In this section, we consider the divisive algorithm which uses the $k$-means objective (with $k = 2$) when choosing how to split clusters. Normally, the $k$-means objective concerns the distances between points and their cluster center: $\min \sum_{i=1}^{k} \sum_{x \in S_i} ||x - \mu_i||^2$. However, it is known that this can be rewritten as a sum over intra-cluster distances: $\min \sum_{i=1}^{k} \frac{1}{|S_i|} \sum_{x,y \in S_i} ||x-y||^2$ [ABC$^+$15]. In other words, when splitting a cluster into two sets $A$ and $B$, the algorithm minimizes $\frac{1}{|A|} \sum_{a,a' \in A} ||a - a'||^2 + \frac{1}{B} \sum_{b,b' \in B} ||b - b'||^2$. At first glance, this appears to almost capture split-$rev_G(A, B)$; the key difference is that the summation has been scaled down by a factor of $|A||B|$. Of course, it also involves minimization over squared distances instead of maximization over similarity weights. We show that the divisive algorithm which splits clusters by the natural $k$-means similarity objective, namely $\max \frac{1}{|A|} \sum_{a,a' \in A} w_{aa'} + \frac{1}{|B|} \sum_{b,b' \in B} w_{bb'}$, is not a good approximation to the optimal hierarchical clustering.

**Lemma 4.1.** *There exists a graph $G = (V, E)$ with nonnegative edge weights $w : E \rightarrow \mathbb{R}^{\geq 0}$, such that if the hierarchical clustering $T^*$ is a maximizer of $rev_G(\cdot)$ and $T$ is the hierarchical clustering returned by the divisive algorithm which, splits clusters by the $k$-means similarity objective, $rev_G(T) \leq \frac{1}{\Omega(\sqrt{n})} rev_G(T^*)$.*

*Proof.* The plan is to exploit the fact that $k$-means is optimizing an objective function which differs from the actual split revenue by a factor of $|A||B|$.

We use almost the same group as in the lower bound against Average Linkage, except that the weight of the edge between $u$ and $v$ is $\sqrt{n}$. There are still unit weight edges between $u$ and $\frac{n}{2} - 1$ other nodes and unit weight edges between $v$ and the remaining $\frac{n}{2} - 1$ nodes. See Figure 1 for the structure of this graph. The key claim is that Divisive $k$-means will begin by separating $u$ and $v$ from all other nodes.

It is easy to see that this split scores a value of $\frac{1}{2}\sqrt{n}$ under our alternate $k$-means objective function. Why does no other split score better? Well, any other split can either keep $u$ and $v$ together or separate them. If the split keeps the two together along with $k$ other nodes, then it scores at most $\frac{1}{k+2}[\sqrt{n} + k] \leq \frac{\sqrt{n}}{k+2} + 1$, which is less than $\frac{1}{2}\sqrt{n}$ if $\sqrt{n} > 6$. If the split separates the two, then it scores at most 2, since at best each side can be a tree of weight one edges and hence has fewer edges than nodes.

Now that we have established our key claim, it is easy to see that Divisive $k$-means is done scoring on this graph, since it must next cut the edge $uv$ and the other larger cluster has no edges in it. Hence Divisive $k$-means will score $\sqrt{n}(n - 2)$ on this graph.

As before, the optimal clustering may merge $u$ with its other neighbors first and $v$ with its other neighbors first, scoring a revenue gain of $2\left[(n-2)+(n-3)+\cdots+(n/2)\right] = \frac{3}{4}n^2 - O(n)$. There is a $\Omega(\sqrt{n})$ gap between these revenues, completing the proof. $\qquad\square$

## 5 Divisive Local-Search

In this section, we develop a simple local search algorithm and bound its approximation ratio. The local search algorithm takes as input a cluster $C$ and divides it into two clusters $A$ and $B$ to optimize a local objective: the split revenue. In particular, initially $A = B = \emptyset$. Each node in $C$ is added to $A$ or $B$ uniformly at random.

Local search is run by moving individual nodes between $A$ and $B$. In a step, any point $i \in A$ (resp. $B$) is added to $B$ (resp. $A$) if $\sum_{j,l\in A;j,l\neq i} w_{j,l} + (|A| - 1)\sum_{j\in B} w_{i,j} > \sum_{j,l\in B} w_{j,l} + |B|\sum_{j\in A,j\neq i} w_{i,j}$ (resp. $\sum_{j,l\in B;j,l\neq i} w_{j,l} + (|B| - 1)\sum_{j\in A} w_{i,j} > \sum_{j,l\in A} w_{j,l} + |A|\sum_{j\in B,j\neq i} w_{i,j}$). This states that a point is moved to another set if the objective increases. The algorithm performs these local moves until there is no node that can be moved to improve the objective.

**Data:** Vertices $V$, weights $w : E \to \mathbb{R}^{\geq 0}$
Initialize clusters $\mathcal{C} \leftarrow \{V\}$;
**while** *some cluster $C \in \mathcal{C}$ has more than one vertex* **do**
    Let $A, B$ be a uniformly random 2-partition of $C$;
    Run local search on $A, B$ to maximize $|B|\sum_{a,a'\in A} w_{aa'} + |A|\sum_{b,b'\in B} w_{bb'}$, considering just
      moving a single node;
    Set $\mathcal{C} \leftarrow \mathcal{C} \cup \{A, B\} \setminus \{C\}$;
**end**

**Algorithm 2:** Divisive Local-Search

In the following theorem, we show that the algorithm is arbitrarily close to a $\frac{1}{3}$ approximation.

**Theorem 5.1.** *Consider a graph $G = (V, E)$ with nonnegative edge weights $w : E \to \mathbb{R}^{\geq 0}$. Let the hierarchical clustering $T^*$ be the optimal solution of $rev_G(\cdot)$ and let $T$ be the hierarchical clustering returned by Algorithm 2. Then, $rev_G(T) \geq \frac{(n-6)}{(n-2)}\frac{1}{3} rev_G(T^*)$.*

*Proof.* Since we know that $rev_G(T^*) \leq (n-2)\sum_{ij} w_{ij}$, it suffices to show that $rev_G(T) \geq \frac{1}{3}(n-2)\sum_{ij} w_{ij}$. We do this by considering possible local moves at every step.

Consider any step of the algorithm and suppose the algorithm decides to partition a cluster into $A, B$. As stated in the algorithm, its local search objective value is $OBJ = |B|\sum_{a,a'\in A} w_{aa'} + |A|\sum_{b,b'\in B} w_{bb'}$. Assume without loss of generality that $|B| \geq |A|$, and consider the expected local search objective $OBJ'$ value for moving a random node from $B$ to $A$. Note that the new local search objective value is at most what the algorithm obtained, i.e. $OBJ' \leq OBJ$:

$$
\begin{aligned}
\mathrm{E}[OBJ'] &= (|B| - 1)\left[\sum_{a,a'\in A} w_{aa'} + \frac{1}{|B|}\sum_{a\in A,b\in B} w_{ab}\right] + (|A| + 1)\left[\frac{\binom{|B|-1}{2}}{\binom{|B|}{2}}\sum_{b,b'\in B} w_{bb'}\right] \\
&= (|B| - 1)\left[\sum_{a,a'\in A} w_{aa'} + \frac{1}{|B|}\sum_{a\in A,b\in B} w_{ab}\right] + (|A| + 1)\left[\frac{|B| - 2}{|B|}\sum_{b,b'\in B} w_{bb'}\right] \\
&= (|B| - 1)\sum_{a,a'\in A} w_{aa'} + \frac{|B| - 1}{|B|}\sum_{a\in A,b\in B} w_{ab} + (|A| + 1)\left[(1 - \frac{2}{|B|})\sum_{b,b'\in B} w_{bb'}\right] \\
&= OBJ - \sum_{a,a'\in A} w_{aa'} + \frac{|B| - 1}{|B|}\sum_{a\in A,b\in B} w_{ab} + (-\frac{2|A|}{|B|} + 1 - \frac{2}{|B|})\sum_{b,b'\in B} w_{bb'}
\end{aligned}
$$

But since there are no improving moves we know the following.

$$0 \geq \mathrm{E}[OBJ'] - OBJ = -\sum_{a,a' \in A} w_{aa'} + \frac{|B|-1}{|B|} \sum_{a \in A, b \in B} w_{ab} - \frac{2|A| - |B| + 2}{|B|} \sum_{b,b' \in B} w_{bb'}$$

Rearranging terms and multiplying by $|B|$ yields the following.

$$(|B|-1) \sum_{a \in A, b \in B} w_{ab} \leq |B| \sum_{a,a' \in A} w_{aa'} + (2|A| - |B| + 2) \sum_{b,b' \in B} w_{bb'}$$

We now consider three cases. Either (i) $|B| \geq |A| + 2$, (ii) $|B| = |A| + 1$, or (iii) $|B| = |A|$. Case (i) is straightforward:

$$\left( \frac{|B|-1}{|A|+|B|} \right) \text{split-cost}_G(A,B) \leq \text{split-rev}_G(A,B)$$

$$\frac{1}{2} \text{split-cost}_G(A,B) \leq \text{split-rev}_G(A,B)$$

In case (ii), we use the fact that $(x+2)(x-2) \leq (x+1)(x-1)$ to simplify:

$$\left( \frac{|B|-1}{|A|+|B|} \right) \text{split-cost}_G(A,B) \leq \left( \frac{|A|+1}{|A|} \right) \text{split-rev}_G(A,B)$$

$$\left( \frac{|B|-1}{|A|+|B|} \right) \text{split-cost}_G(A,B) \leq \left( \frac{|B|+2}{|B|+1} \right) \text{split-rev}_G(A,B)$$

$$\left( \frac{|B|+1}{|B|+2} \right) \left( \frac{|B|-1}{|A|+|B|} \right) \text{split-cost}_G(A,B) \leq \text{split-rev}_G(A,B)$$

$$\left( \frac{|B|-2}{|A|+|B|} \right) \text{split-cost}_G(A,B) \leq \text{split-rev}_G(A,B)$$

$$\left( \frac{1}{2} - \frac{1.5}{|A|+|B|} \right) \text{split-cost}_G(A,B) \leq \text{split-rev}_G(A,B)$$

Case (iii) proceeds similarly; we now use the fact that $(x+2)(x-3) \leq (x)(x-1)$ to simplify:

$$\left( \frac{|B|-1}{|A|+|B|} \right) \text{split-cost}_G(A,B) \leq \left( \frac{|A|+2}{|A|} \right) \text{split-rev}_G(A,B)$$

$$\left( \frac{|B|-1}{|A|+|B|} \right) \text{split-cost}_G(A,B) \leq \left( \frac{|B|+2}{|B|} \right) \text{split-rev}_G(A,B)$$

$$\left( \frac{|B|}{|B|+2} \right) \left( \frac{|B|-1}{|A|+|B|} \right) \text{split-cost}_G(A,B) \leq \text{split-rev}_G(A,B)$$

$$\left( \frac{|B|-3}{|A|+|B|} \right) \text{split-cost}_G(A,B) \leq \text{split-rev}_G(A,B)$$

$$\left( \frac{1}{2} - \frac{3}{|A|+|B|} \right) \text{split-cost}_G(A,B) \leq \text{split-rev}_G(A,B)$$

Hence we have shown that for each step of our algorithm, the split revenue is at least $\left(\frac{1}{2} - \frac{3}{|A|+|B|}\right)$ times the split cost. We rewrite this inequality and then sum over all iterations:

$$\text{split-rev}_G(A, B) \geq \frac{1}{2}\text{split-cost}_G(A, B) - 3 \sum_{a \in A, b \in B} w_{ab}$$

$$\text{rev}_G(T) \geq \frac{1}{2}\text{cost}_G(T) - 3 \sum_{i,j \in [n]} w_{ij}$$

$$= \frac{1}{2}\left(n \sum_{i,j \in [n]} w_{ij} - \text{rev}_G(T)\right) - 3 \sum_{i,j \in [n]} w_{ij}$$

$$\frac{3}{2}\text{rev}_G(T) \geq \frac{n-6}{2} \sum_{i,j \in [n]} w_{ij}$$

$$\text{rev}_G(T) \geq \frac{n-6}{3} \sum_{i,j \in [n]} w_{ij}$$

This is what we wanted to prove. □

We note that it is possible to improve the loss in terms of $n$ to $\frac{n-4}{n-2}$ by instead considering the local search objective $(|B| - 1) \sum_{a,a' \in A} w_{aa'} + (|A| - 1) \sum_{b,b' \in B} w_{bb'}$.

## 6 Conclusion

One purpose of developing an analytic framework for problems is that it can help clarify and explain our observations from practice. In this case, we have shown that average linkage is a $\frac{1}{3}$-approximation to a particular objective function, and the analysis that does so helps explain what average linkage is optimizing. There is much more to explore in this direction. Are there other objective functions which characterize other hierarchical clustering algorithms? For example, what are bisecting $k$-means, single-linkage, and complete-linkage optimizing for?

An analytic framework can also serve to guide development of new algorithms. How well can this dual objective be approximated? For example, we suspect that average linkage is actually a constant approximation strictly better than $\frac{1}{3}$. Could a smarter algorithm break the $\frac{1}{2}$ threshold? Perhaps the $\frac{1}{2}$ threshold is due to a family of graphs which we do not expect to see in practice. Is there a natural input restriction that would allow for better guarantees?

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
