[Supplementary Material]

# Supplementary Material for Approximation Bounds for Hierarchical Clustering: Average Linkage, Bisecting K-means, and Local Search

**Benjamin Moseley**[*]
Carnegie Mellon University
Pittsburgh, PA 15213, USA
moseleyb@andrew.cmu.edu

**Joshua R. Wang**[†]
Department of Computer Science Stanford University
353 Serra Mall, Stanford, CA 94305, USA
joshua.wang@cs.stanford.edu

## 1   A Lower Bound on Average-Linkage-Clustering

This section is devoted to proving the following Lemma 3.2, a lower bound on the performance of average linkage clustering.

**Proof (of Lemma 3.2):**   Our strategy is to trick Average Linkage into collapsing the entire graph into a star graph, while the optimum hierarchical clustering treats the graph as multiple disjoint star graphs. As a warm-up, consider the following graph, depicted in Figure 1. The graph has two special nodes, $u$ and $v$. There is an edge between $u$ and $v$ of weight $w_{uv} = 1 + \delta$ for some small $\delta > 0$. There are also unit weight edges between $u$ and $\frac{n}{2} - 1$ other nodes, and unit weight edges between $v$ and the remaining $\frac{n}{2} - 1$ nodes.

If this is $G$, then Average Linkage will first merge $u$ and $v$ together, scoring a revenue gain of $(1 + \delta)(n - 2) = O(n)$. After this first merge, all nodes appear identical and it does not matter what order they are merged into cluster $\{u, v\}$. Average Linkage will score an additional revenue gain of $(n - 3) + (n - 4) + \cdots + 1 \leq \frac{1}{2}n^2$. Meanwhile, the optimal clustering may merge $u$ with its other neighbors first and $v$ with its other neighbors first, scoring a revenue gain of $2\left[(n - 2) + (n - 3) + \cdots + (n/2)\right] = \frac{3}{4}n^2 - O(n)$. Since Average Linkage has a final revenue of $\frac{1}{2}n^2 + O(n)$ while OPT has a final revenue of $\frac{3}{4}n^2 - O(n)$, as $n$ grows the approximation ratio approaches $\frac{2}{3}$ from above.

We then improve the ratio to $\frac{1}{2}$ considering a clique on $k$ vertices instead of just $u$ and $v$, and giving each node a neighborhood of $n/k - 1$ other vertices. The general graph is depicted in Figure 1.

In the remaining analysis, we treat $k$ as a constant that is hidden by big-O notation. Average Linkage still greedily merges the clique first, scoring a total revenue gain of:

$$(1 + \delta)\left[(1)(n - 2) + (2)(n - 3) + \cdots + (k - 1)(n - k)\right] \leq n\frac{k^2}{2} = O(n)$$

However, after merging the clique, Average Linkage is in the same situation as before and can only score $\frac{1}{2}n^2$ additional revenue.

---

[*]Benjamin Moseley was supported in part by a Google Research Award, a Yahoo Research Award and NSF Grants CCF-1617724, CCF-1733873 and CCF-1725661. This work was partially done while the author was working at Washington University in St. Louis.

[†]Joshua R. Wang was supported in part by NSF Grant CCF-1524062.

Figure 1: Hard graph for Average Linkage (general $k$ case).

In this modified graph, the optimal hierarchical clustering can merge each clique node with its $\frac{n}{k} - 1$ neighbors before merging the clique nodes with each other. However, doing so means that:

$$\text{rev}_G(T^*) \geq k \left[ (n-2) + (n-3) + \cdots + \left( \frac{k-1}{k}n \right) \right]$$

$$= \frac{k}{2} \left( \frac{2k-1}{k}n - 2 \right) \left( \frac{n}{k} - 1 \right)$$

$$= \left( \frac{2k-1}{2k} \right) n^2 - O(n)$$

Following the same analysis as the previous example, our approximation will approach $\frac{1}{2}$ as $k$ grows to infinity. This completes the proof. ∎

## 2 Random Hierarchical Clustering

In this section, we bound the performance of a random divisive algorithm. In each step, the algorithm is given a cluster and divides the points into two clusters $A$ and $B$ where a point is added in each step uniformly at random. We show that this algorithm is a $\frac{1}{3}$-approximation to our revenue function and further this is tight.

---

**Data:** Vertices $V$, weights $w : E \to \mathbb{R}^{\geq 0}$
Initialize clusters $\mathcal{C} \leftarrow \{V\}$;
**while** *some cluster $C \in \mathcal{C}$ has more than one vertex* **do**
  $\quad$ Let $A, B$ be a uniformly random 2-partition of $C$;
  $\quad$ Set $\mathcal{C} \leftarrow \mathcal{C} \cup \{A, B\} \setminus \{C\}$;
**end**

**Algorithm 1:** Random Hierarchical Clustering

---

**Theorem 2.1.** *Consider a graph $G = (V, E)$ with nonnegative edge weights $w : E \to \mathbb{R}^{\geq 0}$. Let the hierarchical clustering $T^*$ be a maximizer of $rev_G(\cdot)$ and let $T$ be the hierarchical clustering returned by Algorithm 1. Then:*

$$\mathrm{E}[rev_G(T)] \geq \frac{1}{3} rev_G(T^*)$$

*Proof.* We begin by pretending that $A$ or $B$ empty is a valid partition of $C$, and address this detail at the end of the proof. If so, we can generate $A, B$ with the following random process: for each vertex $v \in C$, flip a fair coin to decide if it goes into $A$ or into $B$.

Now, consider an edge $(i, j) \in E$. The algorithm will score a revenue of $w_{ij}|\texttt{nonleaves}(T[i \vee j])|$. Thus, we need to determine the expected value of $|\texttt{nonleaves}(T[i \vee j])|$. How often does one of the $n - 2$ other nodes besides $i$ and $j$ become a nonleaf of $T[i \vee j]$? Fix all all coin flips made for $i$ and let $k \neq i, j$ be a point. The point $k$ will become a nonleaf if $j$ matches more coin flips than $k$ does. The number of matched coin flips is a geometric random variable with parameter $1/2$. There is a $1/2$ chance of matching for zero coin flips, a $1/4$ chance of matching for one coin flip, and so on. Hence the probability of equality is $1/4 + 1/16 + 1/64 + \cdots = 1/3$. By symmetry, the remaining $2/3$ probability is split bewteen $j$ matching for more and $k$ matching for more. Hence each of the other $n - 2$ nodes $k$ has exactly a $1/3$ chance of being a nonleaf. As a result,

$$\mathrm{E}[\mathrm{rev}_G(T)] = \frac{n-2}{3} \sum_{ij} w_{ij} \geq \frac{1}{3}\mathrm{rev}_G(T^*)$$

since it is impossible to have more than $n - 2$ nonleaves.

Finally, we address the possibility of $A$ or $B$ being empty. This is equivalent to a node in $T$ having a single child. In this case, $\mathrm{rev}_G(T)$ is unchanged if we merge the node with that child, since this does not change $\texttt{leaves}(T[i \vee j])$ for any edge $(i, j)$. Hence if $A$ or $B$ is empty we can safely redraw. Hence our random process is equivalent to uniformly drawing over all partitions. This completes the proof. □

We now establish that this is tight.

**Lemma 2.2.** *There exists a graph $G = (V, E)$ with nonnegative edge weights $w : E \to \mathbb{R}^{\geq 0}$, such that if the hierarchical clustering $T^*$ is an optimal solution of $\mathrm{rev}_G(\cdot)$ and $T$ is the hierarchical clustering returned by Algorithm 1,*

$$\mathrm{E}[\mathrm{rev}_G(T)] = \frac{1}{3}\mathrm{rev}_G(T^*)$$

*Proof.* In the proof of Lemma 2.1, we showed that

$$\mathrm{E}[\mathrm{rev}_G(T)] = \frac{n-2}{3} \sum_{ij} w_{ij}.$$

This naturally suggests a tight example: any graph where the optimal hierarchical clustering $T^*$ can capture all edges $(i, j) \in E$ with non-zero weight using only clusters of size 2. In other words, in any graph where the edges form a matching, the bound is tight. □