[Reviews · NeurIPS 2017]

Reviewer 1



The authors analyze the lower bounds on some well known clustering algorithms by deriving their approximation ratios. I enjoyed reading the paper. It is well written, and has flow and clarity. A few minor things: - I don't quite understand the reason for the term 'applied' in the title. - The popularity of the algorithms or their practical usage is determined not just by their performance in terms of time, but also their ability to produce better results (both according to intrinsic properties of the clusters as well as comparison with some external 'gold' standard). So I am not sure whether we can say that k-means is not popular in practice because it has lower performance in terms of a poor bound on its approximation ratio. A comparison of algorithms on the basis of time is more appropriate for this claim. - The paper needs to be proof read for some minor language errors e.g. 'except' instead of expect and 'form' instead of from. - The paper ends rather abruptly. It needs a proper Conclusions section.

Reviewer 2



The paper extends the work of Dasgupta towards defining a theoretical framework for evaluating hierarchical clustering. The paper defines an objective function that is equivalent to that of Dasgupta in the sense that an optimal solution for the cost function defined by Dasgupta will also be optimal for the one defined in this paper and vice versa. The behaviour of approximate solution will differ though because the cost function is of the form D(G) - C(T), where D is a constant (depending on the input G) and C(T) is the cost (defined by Dasgupta) and T is the hierarchical solution. The authors show a constant approximation guarantee w.r.t. their objective function for the popular average linkage algorithm for hierarchical clustering. They complement this approximation guarantee by showing that this approximation guarantee is almost tight. They also show that the decisive algorithm for hierarchical clustering gives poor results w.r.t. their objective function. They also claim to show approximation guarantee for a local search based algorithm. Significance: The results discussed in the paper are non-trivial extensions to the work of Dasgupta. Even though the model is similar the the previous work, it is interesting in the sense that the paper is able to give theoretical guarantees for the average linkage algorithm that is a popular technique for hierarchical clustering. Originality: The paper uses basic set of techniques for analysis and is simple to read. The analysis is original as per my knowledge of the related literature. Clarity: The paper uses simple set of techniques and is simple to read. The authors do a fair job in explaining the results and putting things in context with respect to previous work. The explanations are to the point and simple. There are a few places where more clarity will be appreciated. These are included in the specific comments below. Quality: There is a technical error in line 261. It is not clear how the very first inequality is obtained. Furthermore, the last inequality is not justified. The authors should note that as per the analysis, in case (iii) you assume that |B|<=|A| and not the other way around. Given this, I do not know how to fix these errors in a simple way. Specific comments: 1. Please use proper references [C] can mean [CC17] or [CKMM17] 2. Line 69: “Let leaves …”. I think you mean leaves denote the set of leaves and not number. 3.Line 91: Please correct the equation. The factor of n is missing from the right hand side. 4. Line 219: Figure 1 is missing from the main submission. Please fix this, as the proof is incomplete without the figure which is given in the appendix.